# Squamation and scale morphology at the root of jawed vertebrates

Yajing Wang[1], Min Zhu[1,2,3]*

[1]School of Earth Sciences and Engineering, Nanjing University, Najing, China; [2]Key Laboratory of Vertebrate Evolution and Human Origins of Chinese Academy of Sciences, Institute of Vertebrate Paleontology and Paleoanthropology, Chinese Academy of Sciences, Beijing, China; [3]CAS Center for Excellence in Life and Paleoenvironment, Beijing, China

**Abstract** Placoderms, as the earliest branching jawed vertebrates, are crucial to understanding how the characters of crown gnathostomes comprising Chondrichthyes and Osteichthyes evolved from their stem relatives. Despite the growing knowledge of the anatomy and diversity of placoderms over the past decade, the dermal scales of placoderms are predominantly known from isolated material, either morphologically or histologically, resulting in their squamation being poorly understood. Here we provide a comprehensive description of the squamation and scale morphology of a primitive taxon of Antiarcha (a clade at the root of jawed vertebrates), *Parayunnanolepis xitunensis*, based on the virtual restoration of an articulated specimen by using X-ray computed tomography. Thirteen morphotypes of scales are classified to exhibit how the morphology changes with their position on the body in primitive antiarchs, based on which nine areas of the post-thoracic body are distinguished to show their scale variations in the dorsal, flank, ventral, and caudal lobe regions. In this study, the histological structure of yunnanolepidoid scales is described for the first time based on disarticulated scales from the type locality and horizon of *P. xitunensis*. The results demonstrate that yunnanolepidoid scales are remarkably different from their dermal plates as well as euantiarch scales in lack of a well-developed middle layer. Together, our study reveals that the high regionalization of squamation and the bipartite histological structure of scales might be plesiomorphic for antiarchs, and jawed vertebrates in general.

*For correspondence:
zhumin@ivpp.ac.cn

## Editor's evaluation

This manuscript will be of strong interest to scientists studying the development of early jawed vertebrates, in particular the extent and structure of their dermal skeleton, but it will also interest a broader audience given how it connects modern-day morphological techniques to paleobiology. The authors provide the most complete account to date of the body scales of an antiarch stem-group gnathostome; this is the first work to model in 3-D the entire scale cover of such a fossil fish. The authors show that the body scales are varied in form, regionalized and that they comprise two main tissue layers. Based on this they argue that these conditions are plesiomorphic for antiarchs and the gnathostome crown group.

## Introduction

Jawed vertebrates possess a fully mineralized dermal skeleton that falls into two categories: macromery (large bony plates) and micromery (small scales), with the latter differentiating as a squamation across the animal (*Janvier, 1996*; *Jerve et al., 2017*). The squamation is a highly complex integument that varies both interspecifically and intraspecifically (*Reif, 1985*), and may correspond with ecology

**eLife digest** Many vertebrates have an outer skeleton covering their body. Some, like crocodiles, have large bony plates of armor, while others, like fish, have small slippery scales. The type, shape, and arrangement of these structures can tell scientists a lot about how different species evolved.

Most modern fish are completely covered in scales, but this has not always been the case. Over 400 millions of years ago in the Earth's oceans lived a major group of armored fish called antiarch placoderms which had a combination of bony plates, scales and naked skin. These ancient fish are particularly interesting to scientists because they were one of the first jawed vertebrates to evolve. However, much of what is known about this group has come from isolated materials, which has made it difficult to study the organization and shape of their scales.

To overcome this, Wang and Zhu used a specialized x-ray imaging procedure to create a three-dimensional model of one of the best-preserved antiarch placoderm species, *Parayunnanolepis xitunensis.* The model showed that the fish had thirteen types of scales, found in nine distinct regions on its body.

To better understand the structure of these scales, Wang and Zhu looked at the fossils of other extinct jawed fish which where were found in the region where *P. xitunensis* once lived. The scales of these ancient fish were very different from their bony plates, and from the scales of modern fish.

Understanding the skin armor of ancient fish could help to explain how the scales of modern vertebrates evolved. The next step is to look in more detail at the scales of other placoderms to see how they changed over time.

or lifestyle as well as the body plan (*Ferrón and Botella, 2017*). It also contributes to morphological data that have been used to resolve the interrelationships of early jawed vertebrates (*Choo et al., 2017*; *Qiao et al., 2016*).

Placoderms constitute the crownward part of the stem gnathostomes (*Dupret et al., 2014*; *Qiao et al., 2016*; *Zhu et al., 2009*). They primitively bear extensive dermoskeleton (*Giles et al., 2013*), but appear to have recurrently reduced or lost the dermal scales in later species including many advanced arthrodires, euantiarchs, and ptyctodonts (*Denison, 1978*; *Janvier, 1996*). Since the squamation in placoderms has only been preserved in situ in a few examples (*Goujet, 1973*; *Gross, 1963*; *Hemmings, 1978*; *Ivanov et al., 1996*; *Long and Werdelin, 1986*; *Lyarskaya, 1977*; *Stensiö, 1969*; *Upeniece and Upenieks, 1992*), the large number of isolated placoderm scales are of uncertain affinity, and little is known about the complexity of the placoderm squamation.

Concerning the histology in placoderms, gnathal plates and different categories of the dermoskeleton including scales, dermal bony plates, and fin spines have been extensively studied (*Burrow and Turner, 1999*; *Downs and Donoghue, 2009*; *Giles et al., 2013*; *Jerve et al., 2017*; *Rücklin and Donoghue, 2015*; *Young, 2003*). However, the histology of yunnanolepidoids, the most primitive group of antiarchs at the root of jawed vertebrates (*Brazeau et al., 2020*; *Cui et al., 2019*; *Long et al., 2015*; *Zhu et al., 2016*; *Zhu et al., 2013*), is poorly known, especially for the scales.

*Parayunnanolepis xitunensis* from the Lower Devonian of South China, is known for the most complete preserved yunnanolepidoid fossil (*Figure 1A–C*). As such, it was regularly incorporated in the phylogenetic analyses of early jawed vertebrates as a representative of primitive antiarchs (*Brazeau et al., 2020*; *Chevrinais et al., 2017*; *Dearden et al., 2019*; *Giles et al., 2015*; *Zhu et al., 2021*). The scale morphology of *P. xitunensis* was briefly described (*Zhang et al., 2001*; *Zhu et al., 2012*). Nevertheless, the morphological disparity of scales, which would enhance our understanding of the ancestral conditions of jawed vertebrates, was almost unremarked. In the present study, we classify and describe the scales of *P. xitunensis* and establish a squamation model for primitive antiarchs. The histology of yunnanolepidoid scales is investigated to show its deviations from that of yunnanolepidoid dermal bony plates as well as from that of euantiarch scales.

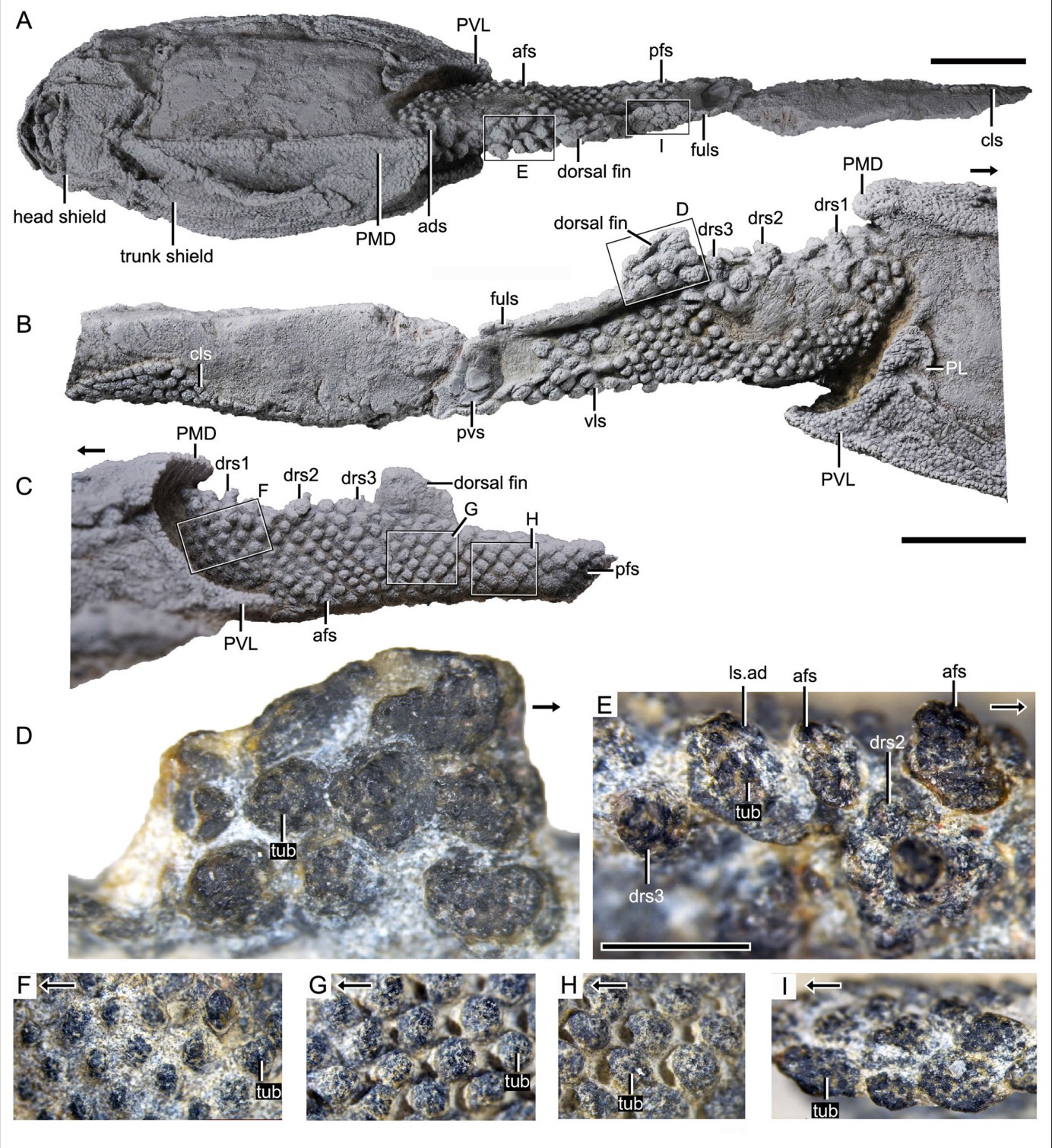

**Figure 1.** Photograph of *Parayunnanolepis xitunensis*, holotype IVPP V11679.1. (**A**) Dorsal view. (**B**) Right lateral view. (**C**) Left lateral view. (**D–I**) Magnified images of rectangles in (**A–C**). (**D**) Dorsal fin in left lateral view. (**E**) Predorsal scales in dorsal view. (**F–H**) Flank scales (fs) in lateral view. (**I**) Postdorsal scales in dorsal view. The black arrow indicates the anterior direction. PL, posterior lateral plate; PVL, posterior ventrolateral plate; other abbreviations see text. A–C share a scale bar of 5 mm, D–I of 1 mm.

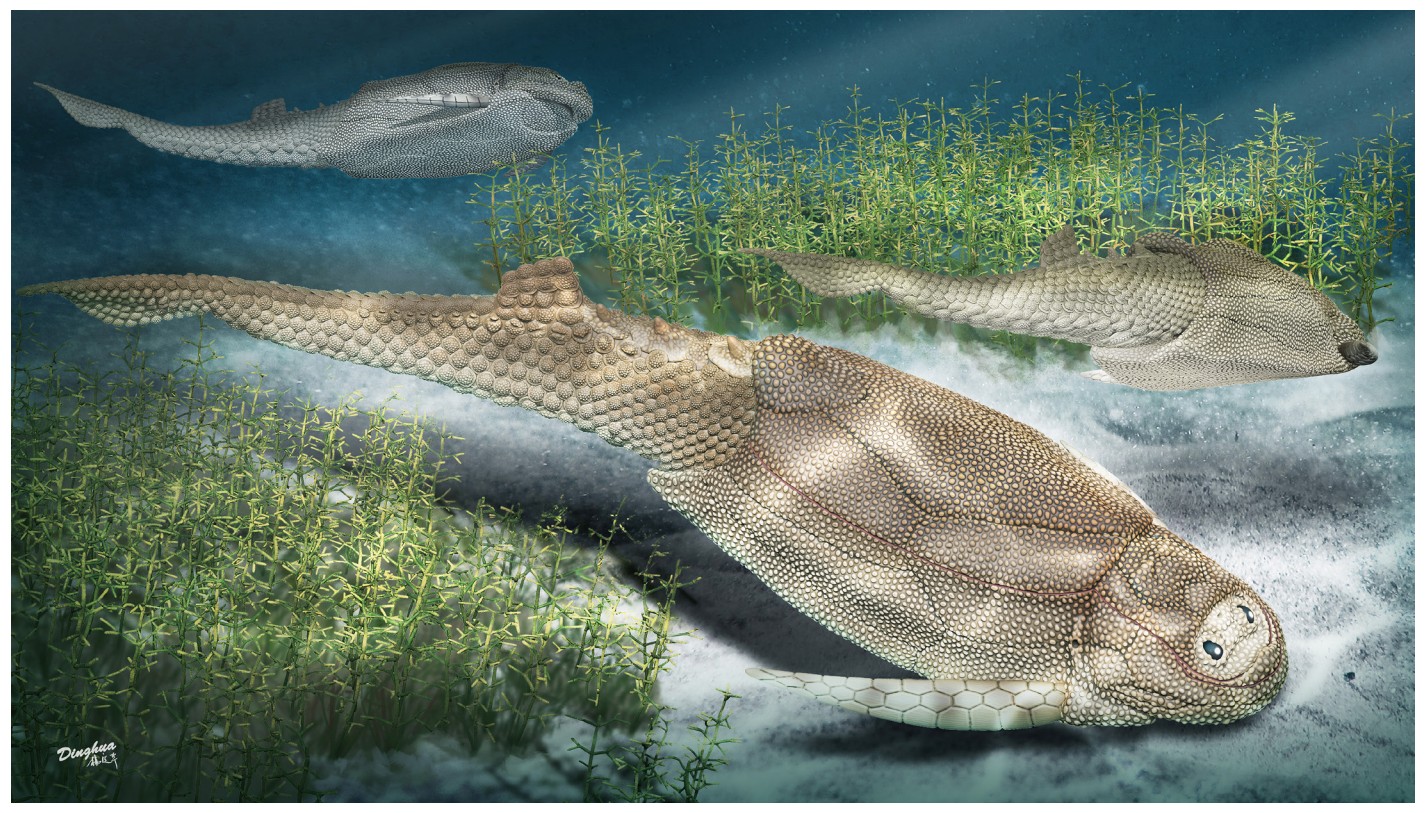

**Figure 2.** Life reconstruction of *Parayunnanolepis xitunensis*, drawn by Dinghua Yang.

## Results

### Scale morphology

The post-thoracic body (*Figure 2*) is completely covered by scales with substantial variability in shape (oval, or rhombic, or polygonal), size (length 0.3–2.6 mm in length, width 0.2–2.3 mm in width, 0.1–0.4 mm in height), overlap relationship (largely overlapping or no overlap), crown/base proportion, and base morphology (bulging, or flat, or concave). Most of the scales are rhombic in shape with their base concave and larger than the crown. Discrete tubercles (tub) ornament the scale crown, and the tubercle size changes with the different scale varieties (*Figure 1D–I*, tub). Basal pores are less developed. Scales are relatively flat, and barely bear the neck structure. The available scales were provisionally assigned to the thirteen morphotypes listed here.

*Morphotype 1*— anterior dorsal scale (ads; *Figure 3D*; *Figure 3—figure supplement 1A–E*). The scale is distinguishable by a thickened crown, rounded anterior margin, and broad outline (width/length, abbreviated as w/l, ranging 1.3–1.6). The depressed field (df; *Figure 3I, J*), that is the area underlying neighboring scales (*Chen et al., 2012*), occupies 28–43% of the scale length. The base is concave under the crown, but convex under the df. The left scale overlaps the right and the back ones. The shape changes from roughly pentagonal at the first row to triangular at the second row, along with the size in length and width getting larger. Following the functional summary for various scale types by *Reif, 1985*, the thickened crown might protect the ads from abrasion in contact with the posterior median dorsal plate (PMD, *Figure 3A*) of the trunk shield.

*Morphotype 2*—dorsal ridge scale (drs1-3; *Figure 3D*; *Figure 3—figure supplement 1F, H, I*). Also termed the 'dorsal scute' or 'median dorsal scale' (*Zhang, 1978*; *Zhu et al., 2012*), it is represented by three large, symmetric, polygonal scales (w/l of about 0.8) behind the ads. Each scale bears a conspicuous bump. The width of the scale decreases backward, with the bump sitting more posteriorly relative to the center of the scale from the first to the third one. The first dorsal ridge scale is featured by a roughly quadrangular outline and a vaulted base. The second one has waved lateral margins and a nearly flat base. The third one, narrow heptagonal in shape, bears developed anterior and lateral

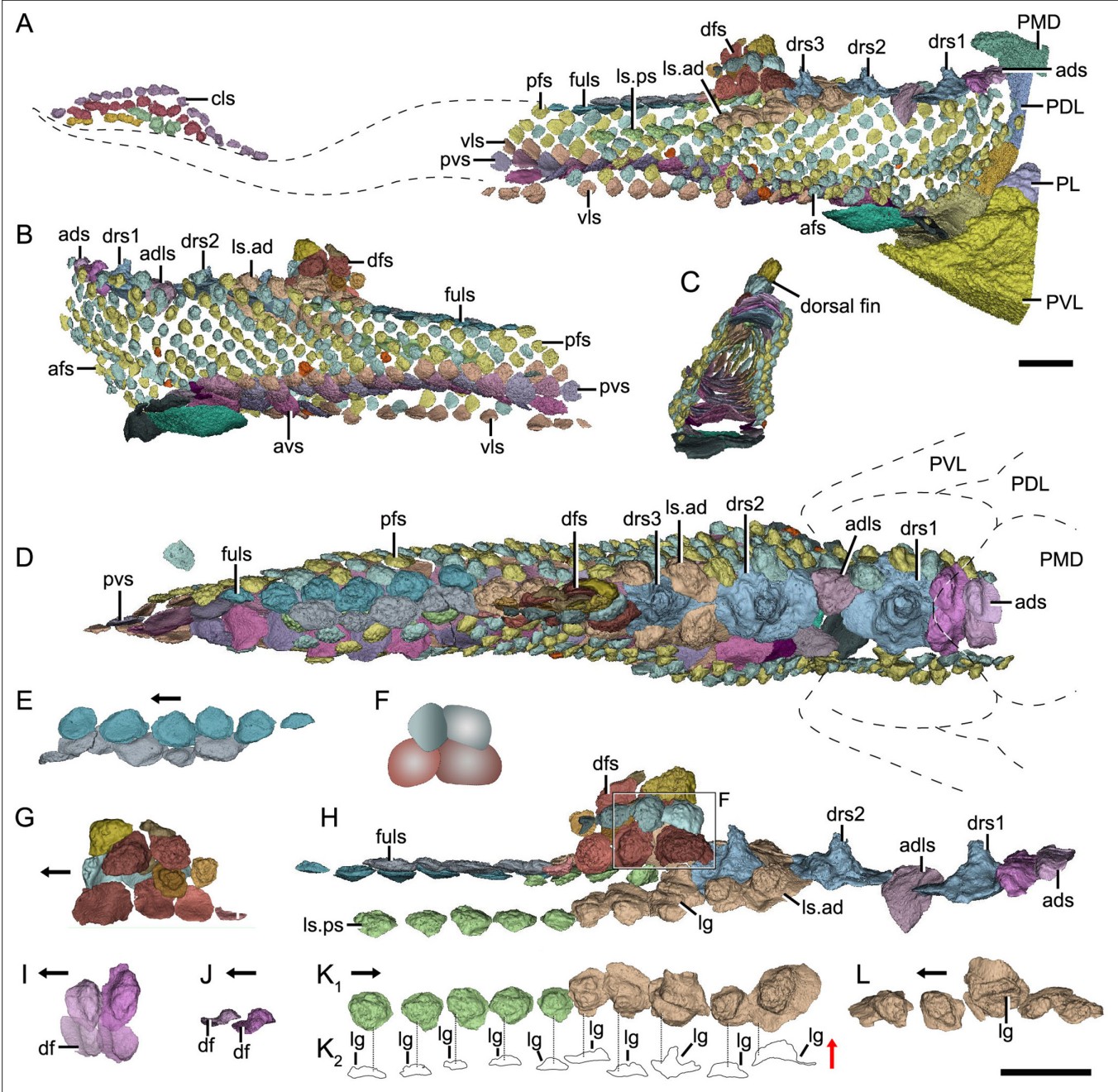

**Figure 3.** Reconstructed squamation of *P.xitunensis* based on CT scanning, IVPP V11679.1. (**A–D**) Squamation in (**A**) right lateral, (**B**) left lateral, (**C**) anterior, and (**D**) dorsal views. (**E**) Posterodorsal fulcral scales in ventral side. (**F**) Interpretative diagram of the dorsal fin scales, showing their contact relationships. (**G**) Dorsal fin scales in left lateral view. (**H**) Dorsal scales in right lateral view. (**I–J**) Anterior dorsal scales in (**I**) dorsal and (**J**) left lateral views. (**K**) Morphotype 6–7 scales on the right side in CT reconstruction (**K₁**) and cross-section diagram (**K₂**), showing the trajectory and profile of the lateral line groove. (**L**) Morphotype 6 scales on the left side. The black and red arrows indicate the anterior and dorsal directions, respectively. Dotted lines indicate the levels of vertical sections shown in (**K₂**). PDL, posterior dorsolateral plate; other abbreviations see text and *Figure 1*. Scale bars equal to 2 mm.

The online version of this article includes the following figure supplement(s) for figure 3:

**Figure supplement 1.** Dorsal scales of *Parayunnanolepis xitunensis* (IVPP V11679.1) based on CT scanning.

**Figure supplement 2.** Dorsal fin scales of *Parayunnanolepis xitunensis* (IVPP V11679.1) based on CT scanning.

corners, and a slender longitudinally directed ridge (dr; *Figure 3—figure supplement 1I*). The base of the third scale is deeply concave beneath the ridge.

*Morphotype 3*—anterior dorsolateral scale (adls; *Figure 3H*; *Figure 3—figure supplement 1G*). Lying between the first and second dorsal ridge scales, the scale has a quadrilateral outline (w/l of 1.17) with a protruding lateral corner (lc; *Figure 3—figure supplement 1G*). A thickening at the turn of the dorsal and ventrolateral laminae (dl, vl; *Figure 3—figure supplement 1G*) strengthens the scale. A df, which is covered by anterior flank scales (afs), occupies almost a half-length of the scale.

*Morphotype 4*—dorsal fin scale (dfs; *Figure 3H*; *Figure 3—figure supplement 2*). The scales are probably arranged into at least three horizontal files on either side of the dorsal fin, with up to four scales on each file (*Figures 1D and 3G*). They decrease progressively in size towards both the dorsal and trailing edges of the fin. In the basal horizontal file, the back scale overlaps the front one, while in the second file, the front scale overlaps the back one (*Figure 3F*), revealing the variability of scale overlap relationships. The first scale in the basal file is oval-shaped, with a cashew-like crown and a strongly concave base; the second and third scales share a round shape and a concave base; the fourth one has a flat base, and is overlapped posteriorly by the morphotype 5 scale. The second file scales can be distinguished by their quadrangular shape, which contrasts with the triangular shape of the first scale in the top file.

*Morphotype 5*—posterodorsal fulcral scale (fuls; *Figure 3D*; *Figure 3—figure supplement 1Q–T*). The scale is rostrocaudally elongated, about 1.2–1.8 times longer than wide, and sub-rectangular in shape. The scales in two files are arranged in a staggered manner (*Figure 1I*), with the left underlying the right. Remarkably, the base of the scale in the left file is concave, while that in the right is convex (*Figure 3E*).

*Morphotype 6*—anterodorsal lateral line scale (ls.ad; *Figure 3H*; *Figure 3—figure supplement 1J–N*). The scales are arranged in an arcuate file around the base of the dorsal fin. They vary in morphology along the body: the first three scales are characterized by a high crown, while the last two by a relatively flat crown; their shape changes from oval to rhombic (w/l of 0.7–1.0). The lateral line canal passes around the crown periphery of the first two scales, but through the crown surface of the following three ones as an open groove (lg; *Figure 3K, L*).

*Morphotype 7*—posterodorsal lateral line scale (ls.ps; *Figure 3H*; *Figure 3—figure supplement 1O, P*). The scale is rhombic in shape with a size intermediate between adjacent flank and fulcral scales. The base is concave and shows a thickening centrally. The scales also carry a lateral line groove that extends from the Morphotype 6 scales and parallels to the longitudinal axis of the body.

*Morphotype 8*—afs (*Figure 4A*; *Figure 4—figure supplement 1A–E, G, I–M*). The scales are oval to sub-rhomboid in shape and form the first 15 (on the right side) or 18 (on the left side) rows (*Figure 4A, B*) on the post-thoracic body. The crown, which is narrower than the base, slightly leans back, and usually beyond the base. The first three rows consist of closely packed scales, while the following rows consist of loosely packed scales. There is a trend of scale size reduction in the top-down direction. A shallow groove on the crowns of three adjacent flank scales in rows 7–9 of the left side represents the anterior elongation of the lateral line on the Morphotype 6 scales (*Figure 4—figure supplement 1B, C*).

*Morphotype 9*—posterior flank scale (pfs; *Figure 4B*; *Figure 4—figure supplement 1F, H*). They are represented by the flank scales lying in rows 16–22 (on the left side) or 18–22 (on the right side), posterior to the trailing edge of the dorsal fin (*Figure 4A, B*). The scales are rectangular (w/l of 0.83) in shape. The flat crown almost covers the base, rarely leaving a df.

*Morphotype 10*—anterior ventral scute (avs; *Figure 5A*; *Figure 5—figure supplement 1A, G*). The ventral scutes in the first to third rows are asymmetric in shape with a straight mesial margin. The crown of the first scute is separated from the base posteriorly by a constricted groove (gr; *Figure 5E–G*). The scute consists of the lateral and ventral laminae (ll.vs, vl.vs; *Figure 5C*). The right scute overlaps the left one in the first two rows, but the left one in the third row overlaps the right one (*Figure 5D, H, I*).

*Morphotype 11*— posterior ventral scute (pvs; *Figure 5D*; *Figure 5—figure supplement 1J–N*). The ventral scutes behind the third row share a staggered arrangement (*Figure 5J, K*). Each scute overlaps the two back ones. The scute is nearly symmetric, with a roughly rhombic shape in most cases. The base is thickened and concave centrally. The upturned posterior margin of the front scute closely articulate with the downturned anterior margin of the back one (*Figure 5B, M*). Compared with Morphotype 10, the lateral lamina in the Morphotype 11 scale is much less developed (*Figure 5C, E*).

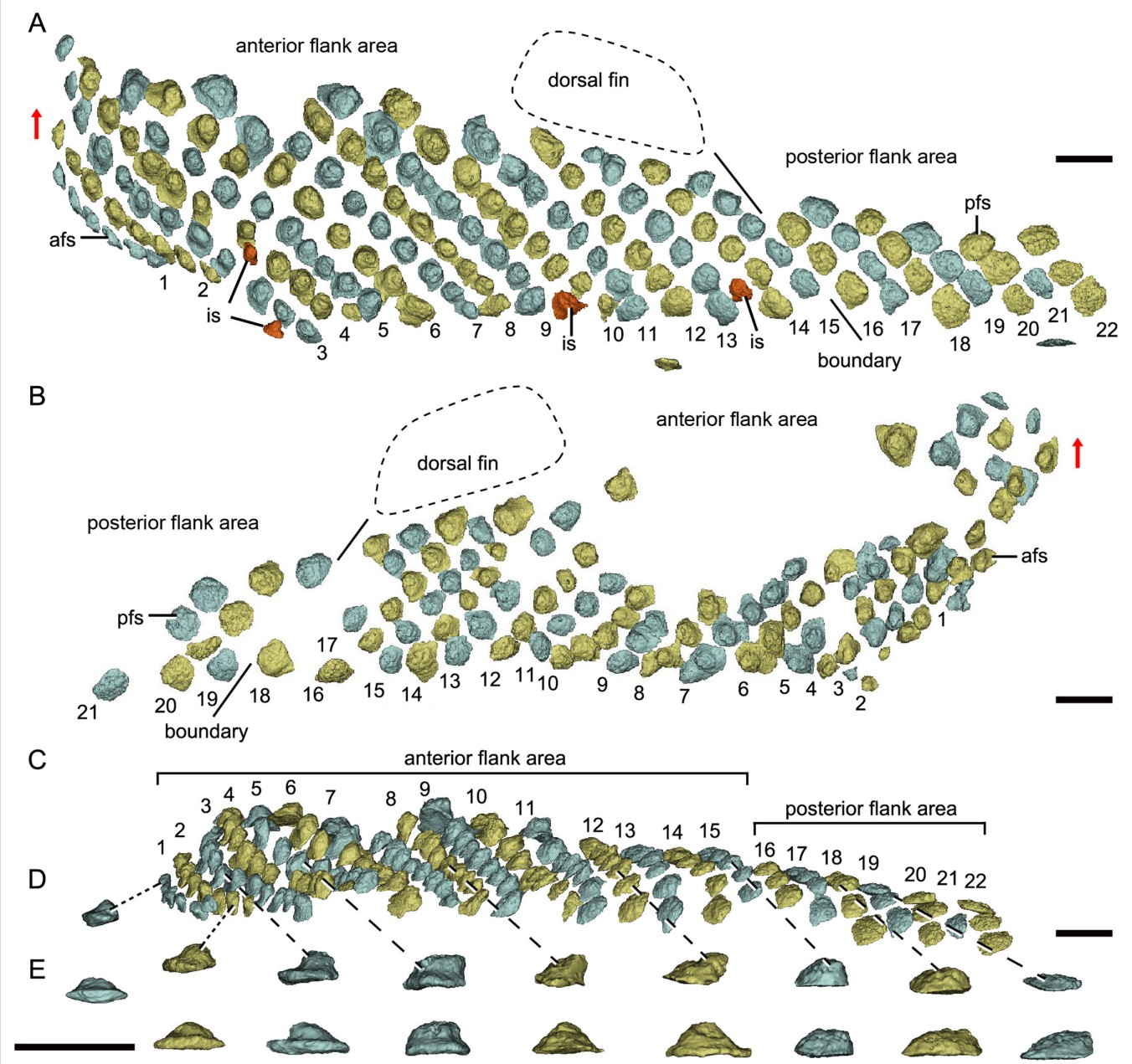

**Figure 4.** Reconstructed flank squamation of *Parayunnanolepis xitunensis* based on CT scanning, holotype IVPP V11679.1. (**A**) Left flank scales in lateral view. (**B**) Right flank scales in lateral view. (**C**) Left flank scales in ventral view. (**D–E**) A series of flank scales in (**D**) ventral and (**E**) anterior views, showing the anterior–posterior gradation of scales. The boundaries between the anterior and posterior flank areas are indicated by a solid line, which represents the extend line of the posterior edge of the dorsal fin. The red scales shown in (**A**) are interspersed scales (is). The red arrow indicates the dorsal direction. Scale bars equal to 1 mm.

The online version of this article includes the following figure supplement(s) for figure 4:

**Figure supplement 1.** Flank scales on the left side of *Parayunnanolepis xitunensis* (IVPP V11679.1) based on CT scanning.

*Morphotype 12*—ventrolateral scale (vls; *Figure 5A*; *Figure 5—figure supplement 1H, I*). The scale is intermediate in size and morphology between the flank scales and ventral scutes. It is characterized by a rhombic shape (w/l of 0.9–1.1), with the mesial corner always at a level in front of the lateral corner. A narrow overlapping occurs between the Morphotype 12 scale and its adjacent ones (*Figure 5D, N*).

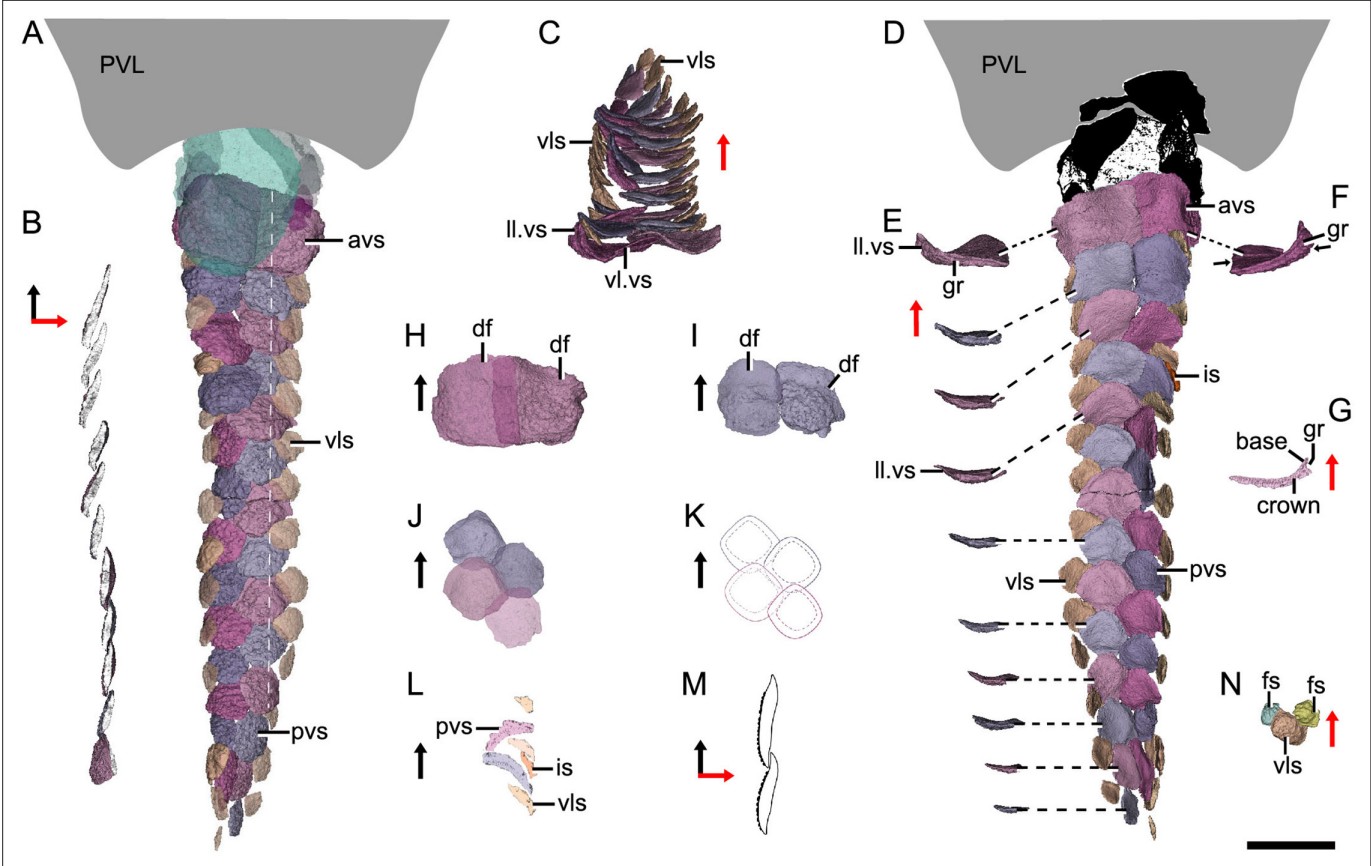

**Figure 5.** Reconstructed ventral squamation of *Parayunnanolepis xitunensis* based on CT scanning, holotype IVPP V11679.1. (**A**) Ventral scales in ventral view. (**B**) Virtual axial section of the ventral scutes at the level indicated by the white dotted line in (**A**). (**C**) Ventral scutes in anterior view. (**D**) Ventral scales in dorsal view. (**E**) A series of ventral scutes in posterior view. (**F**) The first right ventral scute in posterior view. (**G**) Virtual axial section of the scale in (**F**). (**H–I**) Anterior ventral scutes in (**H**) first and (**I**) second rows in ventral views. (**J–K**) Reconstruction and interpretative reconstructions of the posterior ventral scutes in the fourth and fifth rows in ventral view, respectively. (**L**) Virtual sagittal section of the additional scale and its surrounding scales. (**M**) Schematic diagram of the articulated way between ventral scutes in the front and back rows. (**N**) Ventrolateral scale and its surrounding flank scales. The black and red arrows indicate the anterior and dorsal directions, respectively. Scale bar equal to 2 mm.

The online version of this article includes the following figure supplement(s) for figure 5:

**Figure supplement 1.** Ventral scales of *Parayunnanolepis xitunensis* (IVPP V11679.1) based on CT scanning.

*Morphotype 13*—caudal lobe scale (cls; *Figure 3A*). The scales with a small rectangular or square shape cover the hypochordal lobe. They align into four parallel rows in a longitudinal direction.

## Squamation

The post-thoracic body of *P. xitunensis* (*Figure 6*) is about 3.8 mm in width, 6.7 mm in height, and 35.6 mm in length, and occupies approximately 58% of the length overall as noted before (*Zhu et al., 2012*). It is roughly triangular-shaped in the transverse section, and broadest at the level just posterior to the ventral wall of the trunk shield (*Figure 3C, D*). The dorsal fin is positioned at about the first third of the post-thoracic body length (*Figures 1B, 3A, B*), and its preserved part is about 3.0 mm in length, 1.0 mm in width, and 1.6 mm in height.

The scales are packed into oblique dorsoventral rows (flank scales), longitudinal files (dorsal and ventral scales), and curved linear rows (caudal lobe scales) (*Figure 6—figure supplement 1*). According to the scale morphotypes and arrangements, the squamation can be divided into four large regions (dorsal, ventral, flank, and caudal lobe regions), which can be further divided into nine areas (*Figure 6A*).

The dorsal region comprises the predorsal, dorsal fin, posterodorsal, and dorsolateral areas. The squamation in the predorsal area shows the largest disparity of scales (Morphotypes 1–3), which are

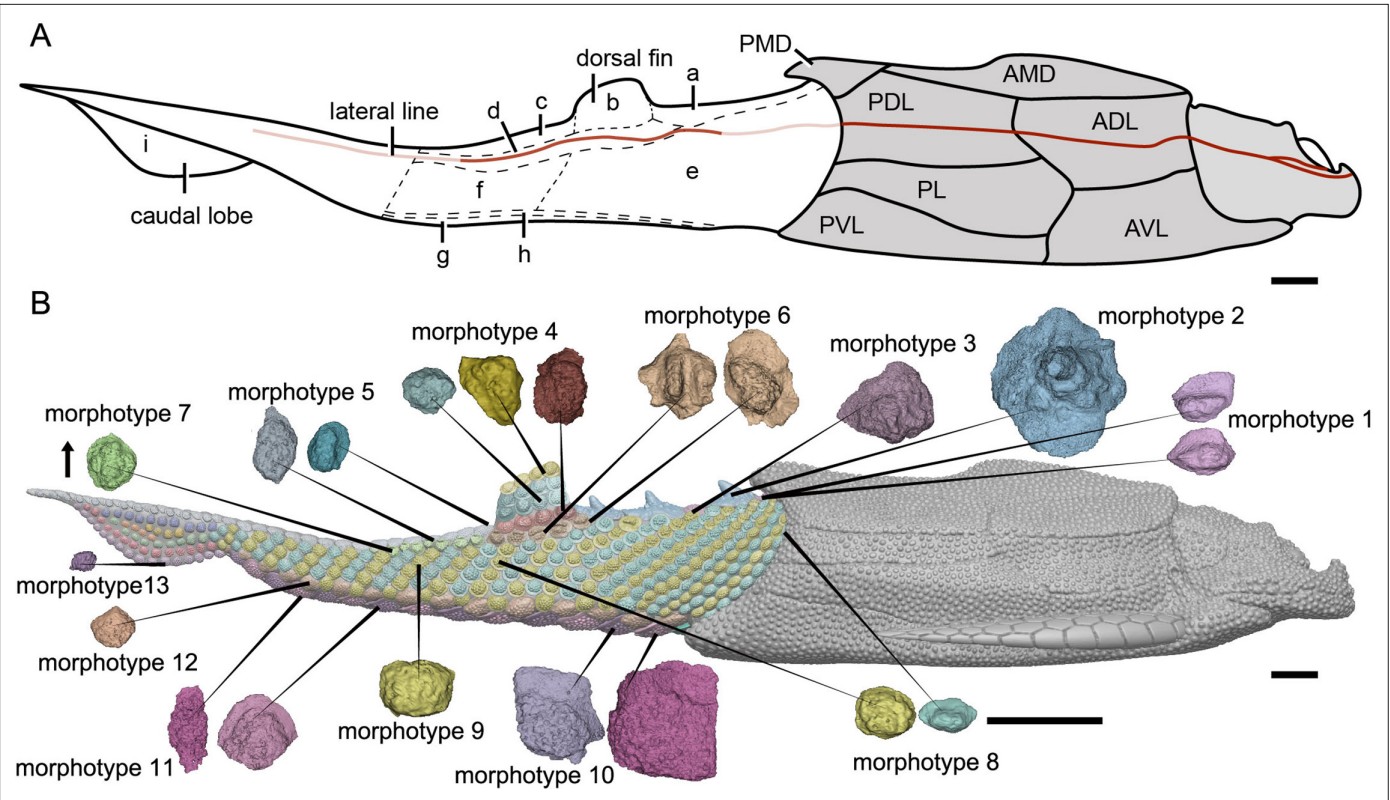

**Figure 6.** Reconstruction of *Parayunnanolepis xitunensis* in lateral view. (**A**) Squamation model showing division of areas. (**B**) 3D model by Dinghua Yang, showing distribution of scale morphotypes. Small letters in (**A**) represent the (**a**) predorsal, (**b**) dorsal fin, (**c**) posterodorsal, (**d**) dorsolateral, (**e**) anterior flank, (**f**) posterior flank, (**g**) ventral, (**h**) ventrolateral and (**i**) caudal lobe areas. Scales from CT reconstruction in (**B**) share the same scale bar and are aligned towards the anterior direction, as indicated by the black arrow. Each morphotype is represented by a typical scale or scales with the most disparity within the morphotype. Deep and light red lines represent the exact and inferred trajectory of the lateral line, respectively. ADL, anterior dorsolateral plate; AMD, anterior median dorsal plate; AVL, anterior ventrolateral plate; other abbreviations see text and ***Figures 1 and 3***. Scale bars equal to 2 mm.

The online version of this article includes the following figure supplement(s) for figure 6:

**Figure supplement 1.** Reconstruction of *Parayunnanolepis xitunensis* by Dinghua Yang.

more heavily imbricated with each other (***Figure 1E***) than the scales in the other areas. Three specialized dorsal ridge scales (Morphotype 2) lie in sequence just behind the ads (Morphotype 1) but in front of the dorsal fin. The first two dorsal ridge scales are separated by a pair of adls (Morphotype 3). The dorsal fin area is completely covered by numerous imbricating scales (Morphotype 4) (***Figure 3G, H***), a feature also known in other placoderms with heavy scale cover. In the posterodorsal area, two files of flat fulcral scales (Morphotype 5) are positioned just behind the dorsal fin. The main lateral line is discernible as an open groove running around or through the crowns of the Morphotype 6–7 scales in the dorsolateral area.

The flank region comprises the anterior and posterior flank areas, which are covered by anterodorsally inclined (c.40–50°) rows of scales (Morphotypes 8–9), with a different row number on either side of the body (***Figure 4A, B***). The disparity of row number might belong to abnormal arrangement of scales caused by suffering trauma (***Gunter, 1948***). The main lateral line in the dorsolateral area extends anteriorly onto the Morphotype 8 scales of the anterior flank area. A small pore opens onto the crown surface in some flank scales (***Figure 4—figure supplement 1G, L***). Remarkably, the squamation in the flank region exhibits a backward gradation of scales in shape (from oval, to rhombic or rectangular), thickness (from high to low crown), and size (from small to large in overall) (***Figure 4C–E***). The size increase from anterior to posterior, might indicate the pfs developed earlier than more anterior ones. However, this size gradation is contrary to that in many other early vertebrates, such as

*Tremataspis* (**Märss et al., 2015**), *Guiyu* (**Cui et al., 2019**), *Mimipiscis* (**Choo, 2012**), and *Gogosardina* (**Choo et al., 2009**).

The ventral region comprises the ventral and ventrolateral areas. The scales in the ventral region are flatter than those in other areas. A file of ventral scutes (Morphotype 10–11) covering the ventral area, is flanked by a file of ventrolateral scales (Morphotype 12) in the ventrolateral area. The squamation in the ventral region also exhibits a backward gradation of scales in shape (from pentagonal to rhombic), size (from large to small), the development of lateral lamina (from well to weakly developed) and overlapping degree (from heavily to slightly overlapped) (*Figure 5A, D*). A few additional scales are interspersed between the regularly arranged ventral scutes and flank scales (*Figure 5D, L*, is), and resembling the surrounding scales in morphotypes. The same condition was also found in the squamation of *Asterolepis* (**Ivanov et al., 1996**).

The caudal lobe region, or the ventral lobe of the caudal peduncle, is covered by tiny scales (Morphotype 13). Being the smallest amongst the squamation (*Figure 6B*), these scales are arranged in linear rows as the tail scales of other antiarchs, such as *Pterichthyodes* (**Hemmings, 1978**).

## Histology of yunnanolepidoid scales

Despite yunnanolepidoid antiarchs displaying abundant fossil records in the Early Devonian sediments of South China (**Burrow et al., 2000**; **Zhao and Zhu, 2010**; **Zhu, 1996**; **Zhu et al., 2000**), the identification of isolated scales has proven difficult as the scale sculpture, mainly composed of round tubercles, is less diagnostic. As such, yunnanolepidoid scales have never been investigated histologically.

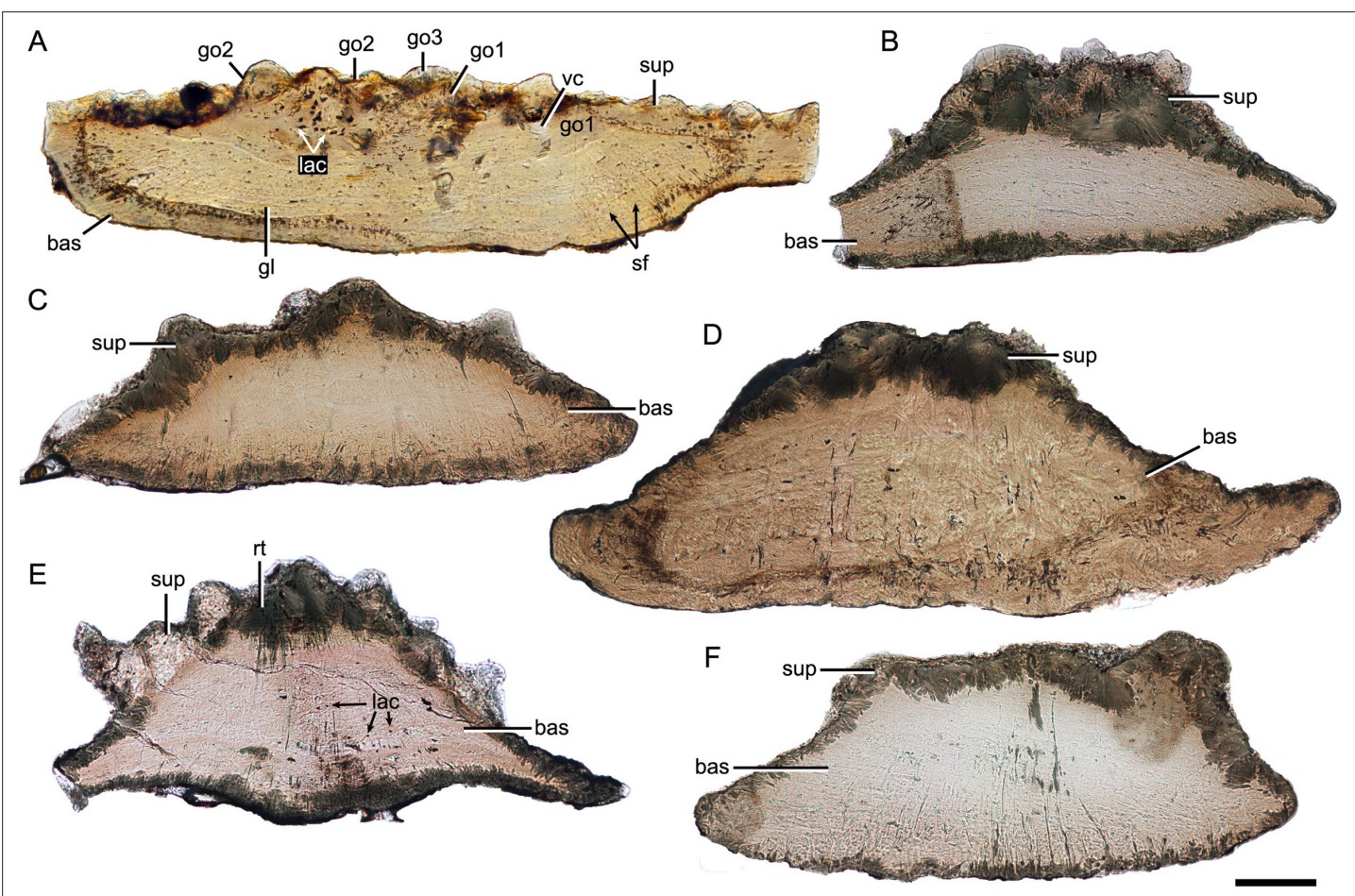

**Figure 7.** Thin sections through yunnanolepidoid scales. (**A**) IVPP V28642. (**B**) IVPP V28643. (**C**) IVPP V28644. (**D**) IVPP V28645. (**E**) IVPP V28646. (**F**) IVPP V29647. Scale bar equal to 100 μm.

The online version of this article includes the following figure supplement(s) for figure 7:

**Figure supplement 1.** Yunnanolepidoid scales.

For the histological research, we selected dozens of isolated placoderm scales from the type locality and horizon of *P. xitunensis*, where yunnanolepidoid antiarchs are the most common placoderm fossils. These isolated scales (*Figure 7—figure supplement 1*) are difficult to be referred to as any definite species, however, they can be assigned to yunnanolepidoids with a reference to the articulated specimen of *P. xitunensis* described here in detail. Displaying various shapes, these scales should come from different species and/or different areas of the post-thoracic body: V28643, V28645, and V28646 can be assigned to Morphotype 8; V28644 and V28647, probably from the posterior flank area, can be referred to Morphotype 9; V28642 can be referred to Morphotype 11. The thin sections through different morphotypes of yunnanolepidoid scales show that the scale uniformly comprises two main divisions: a compact superficial layer, and a thick, lamellar basal layer (sup, bas; *Figure 7*). Thus, the histology of yunnanolepidoid scales is remarkably different from that of their dermal bony plates, which exhibits a three-layered structure including a thick cancellous spongiosa in the middle as in *Yunnanolepis* and *Chuchinolepis* (*Giles et al., 2013*). Significantly, the dermal skeletal differentiation across the dermal shield and scales with respect to the middle layer, is also the case for heterostracans and osteostracans (*Denison, 1947*; *O'Shea et al., 2019*).

The superficial layer of the yunnanolepidoid scale is confined to the tubercular ornament. No apparent dentine tubules could be identified due to recrystallization (rt; *Figure 7E*) of mineralized tissues. However, the dentine tissue was reported in yunnanolepidoid dermal plates (*Giles et al., 2013*; *Young, 2008*), suggesting another potential differentiation across the scale and the dermal plate. The odontodes in the superficial layer display signs of superpositional stacking of at least three generations (go1–3; *Figure 7A*), and the number of superimposed layers of odontodes decreases from the center of the scale to the edge. The younger odontodes cover the older generations of odontodes either completely or incompletely, suggesting a combined areo-superpositional growth (*Ørvig, 1968*) or a compound growth (*Cui et al., 2021*). Polyodontode and the compound growth pattern have been considered to be plesiomorphic in jawed vertebrates (*Cui et al., 2021*; *Qu et al., 2013*). The conditions in yunnanolepidoid scales offer support for these hypotheses. Most of the lacunae are star-shaped with rounded or more elongated cell bodies (lac; *Figure 7A*), and have ramifying canaliculi in all directions. The vasculature might be supplied by occasional ascending canals (vc; *Figure 7A*) from the basal layer, which opened to the surface but was buried later by younger odontodes. In this case, the middle/spongiosa layer may be poorly developed or absent in certain morphotypes. The basal layer is permeated by vertical Sharpey's fibers (sf; *Figure 7A*). These fibers converge towards the center of the scale, at right angles to the lines of the laminae. The scale base shows a series of growth lines (gl; *Figure 7A*).

## Discussion
### Squamation of antiarchs

As described above, *P. xitunensis* exhibits remarkable morphological variability in the scales of nine body areas. Also present in *Romundina* and *Radotina* (*Ørvig, 1975*), another two primitive members of placoderms (*Li et al., 2021*; *Vaškaninová et al., 2020*), the high regionalization of squamation probably represents a primitive character for jawed vertebrates.

In comparison, the squamation regionalization in the ventral and dorsal regions of the body is reduced in euantiarchs (*Figure 8*). Most bothriolepidoids have almost naked bodies, such as *Bothriolepis canadensis*, in which only a small patch of square scales occurs along the lower margin of the tail (*Stensiö, 1948*). The exceptions are *B. gippslandiensis* and *B. cullodenensis* (*Long, 1983*; *Long and Werdelin, 1986*), which have a complete squamation with three regional divisions: the ventrolateral, dorsal fin, and flank areas. *Pterichthyodes milleri* appears to have the highest regional differentiation amongst asterolepidoids, with about six body areas, that is predorsal, dorsal fin, posterodorsal, flank, ventral, and caudal lobe areas (*Hemmings, 1978*). The squamation of *Asterolepis* can be divided into three areas: dorsal fin, dorsal, and flank areas, based on its scale morphotypes (*Ivanov et al., 1995*; *Ivanov et al., 1996*). The squamation of *Remigolepis* can also be divided into three areas: predorsal, posterodorsal, and flank areas (*Johanson, 1997*), or even fewer areas in the species from northern China (*Pan et al., 1987*).

It seems that the squamation is simplified in the aspect of regionalization or disparity in euantiarchs, and further simplified in the clade comprising *Asterolepis* and *Remigolepis*, as their ventral

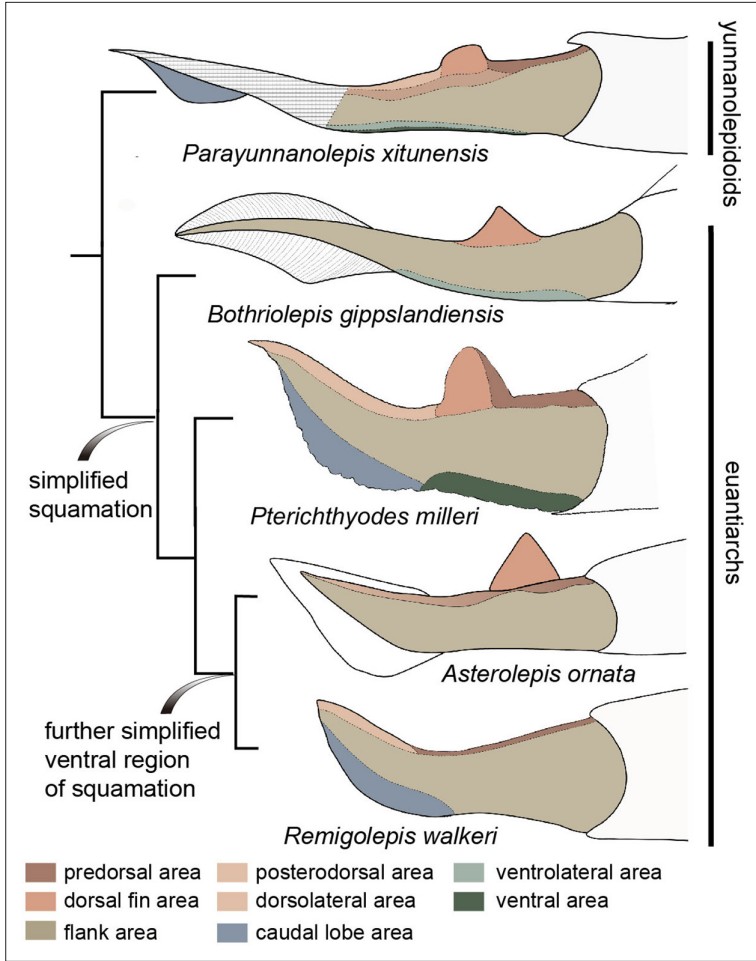

**Figure 8.** Evolution of squamation patterns of the post-thoracic region in antiarchs. Redrawn from the source illustrations of *Long and Werdelin, 1986* for *Bothriolepis gippslandiensis*, *Hemmings, 1978* for *Pterichthyodes milleri*, *Ivanov et al., 1996* for *Asterolepis ornata*, and *Johanson, 1997* for *Remigolepis walkeri*.

scales are not differentiated from the flank scales (*Figure 8*). The simplification of squamation has also occurred in crown sarcopterygians and actinopterygians (*Cui et al., 2019*; *Friedman and Blom, 2006*; *Mondéjar-Fernández and Clément, 2012*), and possibly in crown chondrichthyans (*Ferrón et al., 2018*), suggesting substantial parallelism of squamation in early jawed vertebrates.

## Phylogenetic signals in scale microstructure

The dermal skeleton of heterostracans, osteostracans, placoderms, and early osteichthyans usually consists of three principal layers: a superficial layer composed of dentine and enameloid/enamel; a trabecular/cancellous layer composed of vascular bone; and a compact basal layer composed of lamellar/pseudolamellar bone (*Burrow and Turner, 1999*; *Donoghue and Sansom, 2002*; *Giles et al., 2013*; *Qu et al., 2015b*). This tripartite structure was considered to be plesiomorphic for early jawed vertebrates (*Qu et al., 2015b*; *O'Shea et al., 2019*). However, different dermal units evolve as independent modules (*Qu et al., 2015b*), and vary in histological structure as suggested by the conditions in yunnanolepidoids. In this section, we compare scales of early jawed vertebrates to assess how the variation of diagnostic scale traits including histology and sculpture is correlated with the phylogenetic relationships (*Figure 9*; *Zhu et al., 2016*).

Anaspid, thelodont and galeaspid scales lack an extensive osteonal middly layer (*Burrow et al., 2013*; *Keating and Donoghue, 2016*; *Wang et al., 2005*). Heterostracans have body scales with poorly developed or no middle layer (*Keating et al., 2015*; *Soehn and Wilson, 1990*). In osteostracans or the immediate outgroup of jawed vertebrates, the middle layer with varied development conditions

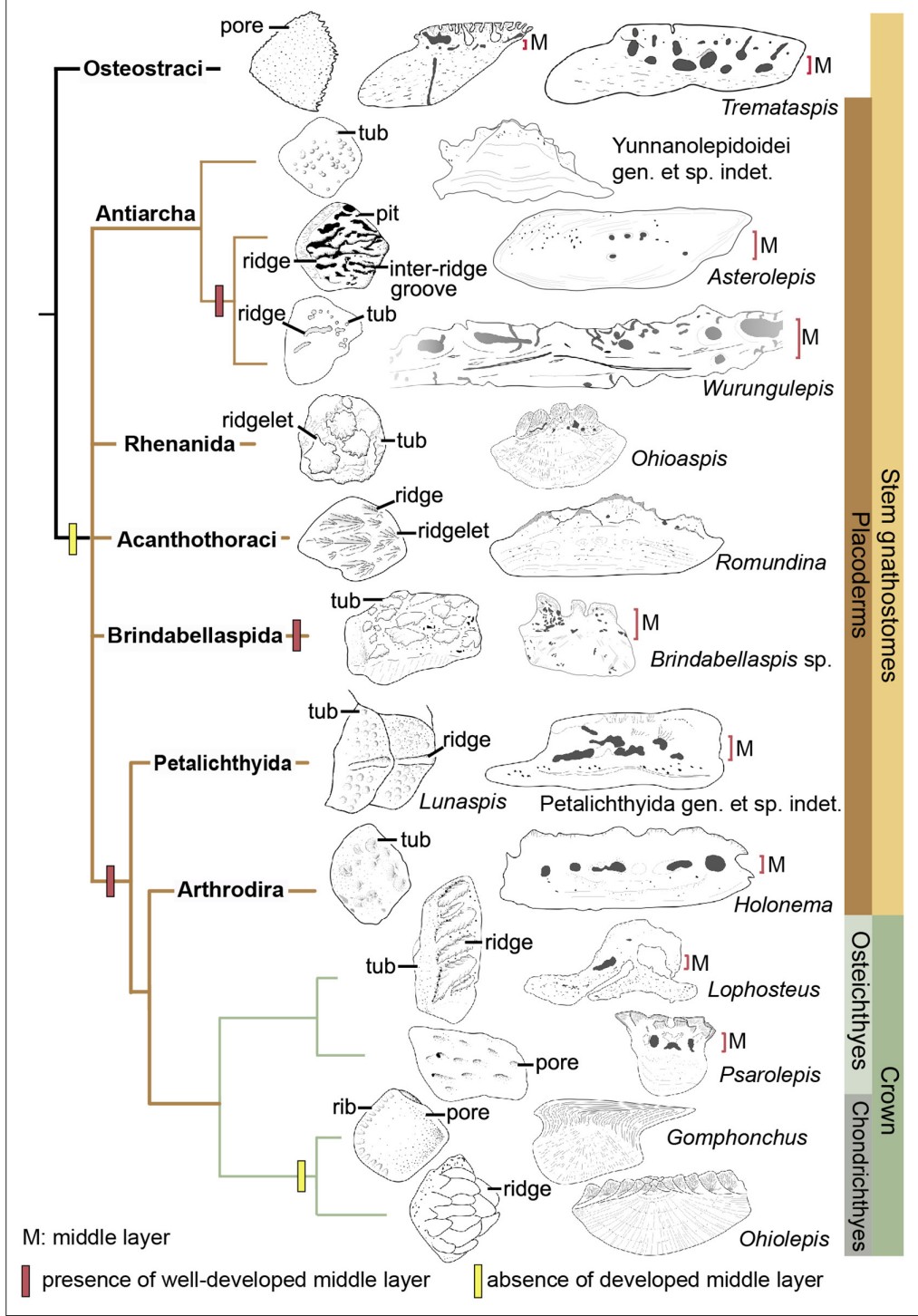

**Figure 9.** Sculpture pattern and histology of scales among gnathostomes. The phylogenetic hypothesis is after *Zhu et al., 2016* as it accords with most of the recent hypotheses about the interrelationships of early jawed vertebrates. Taxon drawings are redrawn from source illustrations of *Märss et al., 2015* and *Qu et al., 2015a* for *Tremataspis*, *Burrow and Turner, 1999* and *Ivanov et al., 1996* for *Asterolepis*, *Burrow and Turner, 1998* and *Young, 1990* for *Wurungulepis*, *Giles et al., 2013* and *Rücklin and Donoghue, 2015* for *Romundina*, *Burrow and Turner, 1999* for *Ohioaspis*, *Burrow and Turner, 1999* for *Lunaspis*, *Burrow and Turner, 1998* for *Brindabellaspis*, *Burrow et al., 2000* for Petalichthyida gen. et sp. indet., *Trinajstic, 1999* for *Holonema*, *Jerve et al., 2016* and *Gross, 1969* for *Lophosteus* , *Qu et al., 2017* for *Psarolepis*, *Gross, 1971* for *Gomphonchus*, and *Gross, 1973* for *Ohiolepis*. Terminology of scale sculpture follows *Märss et al., 2015*. The first column shows scales in dorsal view; the second column shows scales in vertical ground section.

has been reported in tremataspid scales (*O'Shea et al., 2019*; *Qu et al., 2015a*). Ornament on the external surface of the osteostracan scales changes from smooth with pores, to variable shaped tubercles in combination with nodules, ridges, ribs, and inter-ridge grooves (*Märss et al., 2015*).

Yunnanolepidoid scales bear the round tubercle sculpture on the surface and lack the well-developed middle layer, in contrast with that of euantiarchs. The middle partition of scales assigned to *Asterolepis ornata* (*Lyarskaya, 1977*) and *Wurungulepis denisoni* (*Burrow and Turner, 1998*) coincides with the features of the middle layer, such as the 'vascular canals surrounded by osteons'. The *Remigolepis* scale also develops a thick middle layer bearing the upper part of the canal system (*Lukševičs, 1991*). Considering ornamentation, the asterolepidoid scale is much more variable than that of yunnanolepidoid antiarchs; its sculpture changes with the body areas longitudinally from a smooth surface to a combination of nodules, longitudinal crests/ridges, grooves, irregular tubercles, and indistinct pits (*Hemmings, 1978*; *Ivanov et al., 1996*; *Young, 1990*).

In rhenanids, only the histology of *Ohioaspis* scales was investigated (*Burrow and Turner, 1999*). The middle layer is not developed as the main body of vascular canals is buried within the older generations of tubercles (*Gross, 1973*) and thus belongs to the superficial layer. The scales of *Ohioaspis* have stellate tubercles with a multitude of ridgelets around them (*Burrow and Turner, 1999*; *Gross, 1973*), whiles the scales of *Gemuendina* and *Jagorina* are simply ornamented with round tubercles (*Denison, 1978*).

The absence of the spongiosa in the scale was ever perceived as a feature for acanthothoracids (*Burrow, 1996*; *Denison, 1978*). The middle layer is very weakly developed in scales of *Jerulalepis* and radotinids (*Burrow, 1996*; *Stensiö, 1969*). However, different conditions occur in the scales referred to *Murrindalaspis*, *Connemarraspis*, and *Romundina*, which bear a well-developed middle layer in addition to the complex ornamentation composed of stellate, ridged or polygonal tubercles as in the stem osteichthyan *Lophosteus* (*Burrow, 2006*; *Burrow and Turner, 1998*; *Gross, 1969*; *Long and Young, 1988*). Noteworthy is that the middle layer in the scales of *Romundina* exhibits two different conditions: poorly developed in some thin sections (*Giles et al., 2013*; *Rücklin and Donoghue, 2015*; *Smith et al., 2017*), and well-developed in other sections (*Giles et al., 2013*). This histological difference in the *Romundina* scales probably results from individual disparity, as suggested by *Tremataspis mammillata* with reduction of the middle layer posteriorly across the squamation (*O'Shea et al., 2019*).

In brindabellaspids, the middle layer in the *Brindabellaspis* scale is developed (*Burrow and Turner, 1999*), and the ornamentation of the *Brindabellaspis* scale is distinct in consisting of arrowhead-shaped tubercles (*Burrow and Turner, 1998*).

Petalichthyids and arthrodires constitute the successive sister groups of jawed crown gnathostomes, and their scales generally bear a well-developed middle layer (*Burrow et al., 2000*; *Trinajstic, 1999*), with a few exceptions of buchanosteid scales (*Burrow and Turner, 1998*). Concerning the ornamentation, the petalichthyid scales usually bear a diagonal ridge, surrounded by rounded or stellate tubercles (*Burrow et al., 2000*; *Denison, 1978*). The arthrodire scales are largely covered by rounded tubercles, or stellate tubercles that are either surrounded centrally by larger tubercles or combined with longitudinal ridges (*Burrow and Turner, 1999*; *Mark-Kurik and Young, 2003*).

In crown gnathostomes, the middle layer occurs extensively in early osteichthyans, such as *Lophosteus* (*Jerve et al., 2016*) and *Psarolepis* (*Qu et al., 2013*; *Schultze, 2016*), but is widely absent in the chondrichthyan lineage including acanthodians (*Andreev et al., 2016a*; *Andreev et al., 2016b*; *Andreev et al., 2020*; *Burrow et al., 2020*; *Burrow et al., 2016*; *Burrow et al., 2013*; *Chevrinais et al., 2017*; *Hanke and Wilson, 2010*). The ornamentation on dermal scales seems slightly more variable and complex in osteichthyans than that of placoderms, as exampled by the *Andreolepis* and *Lophosteus* scales. *Andreolepis* usually carries parallel or united ridges along with grooves and pores (*Chen et al., 2012*), and *Lophosteus* carries serrated ridges (*Jerve et al., 2016*). The scale surface in the chondrichthyan lineage is often marked by pectinate ornamentation, consisting of raised ridges, grooves, ribs, oriented tubercles or denticulate ornament, occasionally accompanied by foramina.

The osteonal middle layer was hypothesized evolved in a clade consisting of osteostracans plus jawed vertebrates (*Keating and Donoghue, 2016*). Then the middle layer of scales is reduced or absent in the jawed vertebrate node, since the bipartite scale (a thick basal layer +a superficial layer) is more likely to be a common feature for primitive placoderms (*Burrow, 1996*; *Denison, 1978*) with new evidence from yunnanolepidoid scales. The middle layer subsequently independently evolved several times in scales of euantiarchs, brindabellaspids, and a clade comprising more crownward

placoderms including petalichthyids and arthrodires, plus crown gnathostomes, but lost again in the chondrichthyan total group (*Figure 9*). Alternatively, under the recent phylogenetic hypothesis by *Zhu et al., 2021*, the most parsimonious solution is that the well-developed middle layer independently evolved four times in petalichthyids + arthrodires, brindabellaspids, euantiarchs and osteichthyans.

## Conclusions

A comprehensive study on the squamation of *P. xitunensis* using X-ray computed tomography and the histology of disarticulated scales provides the following new features in yunnanolepidoid antiarchs: (1) scales are generally rhombic, thick and show a concave base, but still vary in shape, size, ornamentation, concavity of the base, and overlap relationships in one individual, based on which we classified them into at least thirteen morphotypes for *P. xitunensis*; (2) nine areas of the post-thoracic body are distinguished to show the scale variations in the dorsal, flank, ventral, and caudal lobe regions, revealing the high regionalization of squamation at the root of jawed vertebrates; (3) yunnanolepidoid scales are histologically bipartite, composed of a thin upper superficial layer with multiple generations of tubercles and a thick basal layer.

Thin sections for yunnanolepidoid scales, together with evidence from other gnathostomes suggest that the well-developed middle cancellous layer of scales is primitively reduced or absent in jawed vertebrates, and independently evolved in euantiarchs, brindabellaspids, and a clade uniting petalichthyids, arthrodires, and crown gnathostomes.

## Materials and methods

This study involves the holotype of *P. xitunensis* IVPP V11679.1 and six disarticulated scales IVPP V28642–V28647 (*Figure 7—figure supplement 1*) from Xitun village, Cuifengshan in Qujing city (Yunnan Province). They were collected from the middle part of the Xitun Formation (late Lochkovian) and housed in the Institute of Vertebrate Paleontology and Paleoanthropology.

V11679.1 was scanned using the Nano-CT system (Phoenix x-ray, GE Measurement & Control) at a pixel resolution of 5.99 μm in the Key Laboratory of Vertebrate Evolution and Human Origins of Chinese Academic of Science, Beijing. It was scanned at 90 kV, 100 μA, 1 s exposure time, with a 9 μm focal spot size. V28642–V28647 were processed with the diluted acetic acid and then scanned using the Nano-CT at 80 kV and100 μA, with a 1.85 μm voxel size. Raw data were pre-processed in VG Studio Max 3.3, and reconstructions were performed using Mimics 19.0.

## Acknowledgements

We thank H X Lin (IVPP), Y M Hou (IVPP), and P F Yin (IVPP) for their assistance with CT scanning, and to L T Jia (IVPP) for his assistance with photographs. We also thank L J Peng (Qujing Normal University) for his help in making thin sections and Q M Qu (Xiamen University) for his constructive suggestions. This work was supported by the National Natural Science Foundation of China (42130209), and the Strategic Priority Research Program of the Chinese Academy of Sciences (XDA19050102 and XDB26000000).

## Additional information

### Competing interests

Min Zhu: Reviewing editor, eLife. The other author declares that no competing interests exist.

### Funding

| Funder | Grant reference number | Author |
| --- | --- | --- |
| National Natural Science Foundation of China | 42130209 | Min Zhu |
| Strategic Priority Research Program of the Chinese Academy of Sciences | XDA19050102 | Min Zhu |

| Funder | Grant reference number | Author |
|---|---|---|
| Strategic Priority Research Program of the Chinese Academic of Sciences | XDB26000000 | Min Zhu |

The funders had no role in study design, data collection and interpretation, or the decision to submit the work for publication.

## Author contributions
Yajing Wang, Conceptualization, Data curation, Investigation, Writing – original draft, Writing – review and editing; Min Zhu, Conceptualization, Data curation, Funding acquisition, Methodology, Resources, Supervision, Writing – original draft, Writing – review and editing

## Author ORCIDs
Yajing Wang http://orcid.org/0000-0002-2230-3526
Min Zhu http://orcid.org/0000-0002-4786-0898

## Decision letter and Author response
Decision letter https://doi.org/10.7554/eLife.76661.sa1
Author response https://doi.org/10.7554/eLife.76661.sa2

# Additional files

## Supplementary files
• Transparent reporting form

## Data availability
CT scan data (exported as .mcs files), reconstructions (exported as .stl files), and measurement data generated during this study will be available at https://doi.org/10.5061/dryad.fttdz08vb upon acceptance of the manuscript.

The following dataset was generated:

| Author(s) | Year | Dataset title | Dataset URL | Database and Identifier |
|---|---|---|---|---|
| Zhu M | 2022 | Squamation and scale morphology at the root of jawed vertebrates | https://doi.org/10.5061/dryad.fttdz08vb | Dryad Digital Repository, 10.5061/dryad.fttdz08vb |

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
