## [Editor Report]

This manuscript will be of strong interest to scientists studying the development of early jawed vertebrates, in particular the extent and structure of their dermal skeleton, but it will also interest a broader audience given how it connects modern-day morphological techniques to paleobiology. The authors provide the most complete account to date of the body scales of an antiarch stem-group gnathostome; this is the first work to model in 3-D the entire scale cover of such a fossil fish. The authors show that the body scales are varied in form, regionalized and that they comprise two main tissue layers. Based on this they argue that these conditions are plesiomorphic for antiarchs and the gnathostome crown group.

---

## [Decision Letter]

**Decision letter after peer review:**

Thank you for submitting your article "Squamation and scale morphology at the root of jawed vertebrates" for consideration by *eLife*. Your article has been reviewed by 3 peer reviewers, and the evaluation has been overseen by a Reviewing Editor and George Perry as the Senior Editor. The following individuals involved in the review of your submission have agreed to reveal their identity: Carole J. Burrow (Reviewer #1); Richard Dearden (Reviewer #3).

Essential revisions:

The reviewers were positive overall, but they also passed along some clear and reasonable requests for revision, provided below. These include revising the discussion to be more open to alternative phylogenetic schemes and to compare the data a bit more broadly with jawless gnathostomes. Doing so might make the story a bit less clear but it would help make the manuscript more informative and resistant to future changes in our understanding of phylogenetic topology and so on. You are also asked to consider the size and maturity of the specimen and to ensure that you are providing all data that were used in the paper. We also had some discussion concerning Figure 6a; please consider the difference between a spongiose middle layer and a network of canals in the base of the ornament layer in your interpretation/discussion.

*Reviewer #2 (Recommendations for the authors):*

In general, the article is well-written and provides clear and concise descriptions of the various morphotypes and squamation of the individual specimen; however, there are many sections where there are some grammatical or editorial mistakes. These include:

Lines 32-34: How does this study demonstrate that yunnanolepidoid scales are different from the dermal plates of antiarchs, but also include yunnanolepidoids? The end of this sentence is a bit confusing.

Line 45: I don't think the word "perfectly" should be used here as Reif, 1985 is just suggesting that different dermal denticles in sharks might correspond with different functions due to ecology or lifestyle. There hasn't been any explicit research showing evidence of this.

Lines 59-60: This sentence is confusing. The gnathal plates on placoderms can be extensively studied, not the other way around.

Lines 68-69: The way the sentence is written currently suggests that P. xitunensis is known for preserving squamation. I think here you mean the "fossil" or "preserved specimen".

Lines 81-83: This sentence doesn't make sense.

Line 103: What do you mean by "the size getting larger"? Increase in length, width?

Line 128: What changes the outline?

Line 430: How big was this specimen? There is some evidence that dermal scales in sharks change throughout ontogeny so knowing the size of this specimen may help elucidate squamation growth if future individuals are recovered or examined.

There are also a few other formatting or content errors I would like to point out:

Line 43: Jerve et al., 2017 isn't the actual article that differentiates the dermal skeleton into two different categories. I think this should be "Janvier P. Early Vertebrates. 1996".

Lines 54-56: Why are the citations here italicized?

Line 273: Do you mean "never" not ever? This is also done in Lines 318 and 393.

Line 308: Could you give some examples of crown sarcopterygians and actinopterygians that you are referencing here?

Line 321: Should be "In this section" not "In this part".

Line 442: All of these abbreviations should be used in the text when the figure is referred to.

There were also some inconsistencies and issues with the figures. In general, there were some parts of the figures that weren't mentioned at all in the text. Make sure to go back through the manuscript and correct these. Additionally, you should also include the small-lettered abbreviations within the text in the parentheses with the Figure numbers (as was done in Line 144). These abbreviations are not included in the figure captions and I think it would lend to a more easily read manuscript if they were included within the text. There are some other things corresponding to the reference of figures within the text that I would like to point out:

Line 87: Is the post-thoracic body indicated by the white region in Figure 1J? If so, this should be referenced here.

Line 93: Tubercles should be labeled in Figure 1.

Line 179: The lateral lamina is referenced with Figure 4C, E, but there is no lateral lamina in this figure.

Line 199: I think you mean supplemental figure 6, not 5.

Line 326: The manuscript says that in tremataspid scales, the middle layer is well described; however, this is not depicted in Figure 8.

Lines 353-354: Again, the text says that Romundina bear a well-developed middle layer; however, this is not shown in Figure 8.

Line 401: Ornamentation is mentioned here but is not shown or labeled in Figure 8.

Line 426: This should actually be supplemental figure 5, not 6.

Generally, the figures are really great and have a lot of information and detail that add to the manuscript; however, I think the authors should take some time to make sure the images can stand alone without needing the manuscript text for explanation. Additionally, there are some things specifically about the Figures themselves that I think could be changed to increase the readability of the manuscript:

Line 713: Should read "(D-I) magnified images of rectangles in (A-C)".

Figure 3 A and B: The lines that are separating the posterior flank area from the anterior flank area should be better described or labeled on the image. Also, the authors should point out here in the caption what the orange scales are.

Figure 7: The title should be "Evolution of squamation patterns of the caudal region in antiarchs".

Figure 8: You should include what the view or plane is of the scales (or the two columns) we are looking at. Is the first column dorsal or lateral? Additionally, why is there a "?" in front of the M in the first row? It might be better if you change the colors of the presence and absence of the middle layer because currently, the color of the Middle layer bracket in column 2 is the same color as the bar representing a lack of a middle layer in the phylogeny. It should be consistent.

Figure supplement 2: Line 802, should the Dorsal view letters be subscript "2" and not "1"?

Figure supplement 3: How can the E scales be both anterior and posterior flank scales? Also, the label "H" is confusing. What does that point to?

Figure supplement 5: Which morphotype are these scales? Are they all the same? Also, which plane are the rows showing?

*Reviewer #3 (Recommendations for the authors):*

I enjoyed reading this article, it's great to see people working on the posterior half of placoderms as opposed to their more popular business ends. I think that the articulated squamations of these things have untapped potential as a source of characters and information on vertebrate evolution which work like that presented here is necessary to access. Moreover, this data is linked to an animal commonly used in phylogenetic matrices targeting early vertebrates and I'm sure will be incorporated into future such work.

The introduction outlines necessary background information clearly and succinctly. In terms of the analysis, CT scanning and thin sections are both very appropriate methods for investigating this thing's squamation, and the interpretation of this and the morphotypes proposed by the authors seem sensible. I think the discussion could range a little more broadly in terms of comparing the scales to those of other early vertebrates: more details are given below.

My only issue with the interpretation of the data is in Figure 6A, where it seems to be there are some canals passing through the section. Why do the authors not consider this to be a middle layer? Even if they are confident it is not, this is worth discussing in the text. It seems to me that it's not impossible that the presence/absence of a middle layer can vary in a single taxon, depending on the size or the ornamentation of scales in different places. Dearden et al. 2021 (p22) suggest that "putative cancellous layer in placoderm scales could be an overinterpretation of the canals above the base in crown tubercles of placoderm scales (which also occur in stem-group chondrichthyan scales), rather than corresponding to the spongiosa or trabecular layer in placoderm dermal bone." which could be the case here?

Given the tendency toward turmoil in stem-gnathostome relationships, I also think that it would benefit from considering alternative phylogenetic hypotheses for the phylogenetic position of antiarchs in the gnathostome stem group. While it's true that usually they are recovered in a fairly early-branching position in the gnathostome stem group, a recent analysis by Zhu et al. recovered them in quite a crownwards position. This is worth considering as an alternative when hypothesising about the evolution of vertebrate squamations. Or, if the authors do not agree with the phylogenetic proposition of that study, they should state why.

Also, as noted in the figures section, the scheme of gain/absence in figure 8 is equally parsimonious to one where the middle layer is lost 4 times. What is the basis for this character optimisation?

Another general point is that the authors repeatedly refer to mandibulate gnathostomes (i.e. gnathostomes with jaws) as "gnathostomes", for example when describing osteostracans as the outgroup to jawed vertebrates. Given that a total-group framework, which leads to numerous jawless vertebrates including osteostracans in the gnathostome stem group, has been widely adopted in early vertebrate palaeontology this can read a bit confusingly. I recommend that throughout, the authors are more terminologically precise: the use of e.g. "mandibulate stem-gnathostomes" etc might help.

This dataset provides a good opportunity to compare the squamation of Parayunnanolepis more broadly than the authors do at present. If antiarchs are indeed the earliest branching mandibulate jawed vertebrates then it is worth comparing their squamation beyond the immediate outgroup (osteostracans) to articulated squamations in heterostracans, theodonts, anaspids, galeaspids (if any exist?).

My line by line comments follow:

Abstract

Line 16: "the most basal jawed vertebrates". Potentially misleading as the word basal is ill-defined. Moreover, even if it is taken to mean the earliest branching jawed vertebrates, "ostracoderms" are the most "basal" jawed vertebrates. Recommend rewriting using terminology like "earliest branching mandibulate jawed vertebrates".

Line 42: "as a homogenous squamation across the animal". This isn't strictly speaking true: many acanthodians have surprisingly heterogenous squamations with (e.g. Ptomacanthus Brazeau 2012, Vernicomacanthus Dearden et al. 2021), as do living chondrichthyans (e.g. Naylor et al. 2021) All would be considered micromeric.

Line 45 Does anything in nature correspond "perfectly"?

Line 49 Like basal, "upper" is terminologically unclear. Using more precise terminology like "more crownwards grade" or "a grade with a closer relationship to the crown group" would be clearer.

Line 64 "at the root of jawed vertebrates" See comments on line 16. Also, see the general comment above on alternative hypotheses of relationships.

Line 86 I find the authors' scale morphotypes convincing, but the paper would benefit from an additional figure showing all 13 morphotypes side by side. This will help readers get a quick overview and sense of the variation and morphology of the morphotypes.

Line 90. I think in the summary descriptions it is also worth stating what characters that readers might expect in early vertebrate scales are absent. For example, are there any basal pores or neck canals? I understand that there are not, but it is worth stating their absence as they are potentially important characters of early vertebrate body scales.

Line 125 Morphotype 4 could do with a more detailed description of the morphology to bring it in line with the other morphotype descriptions.

Line 204. “…predorsal area shows the large disparity…” Should read “largest”?

Line 214. It would be useful to figure a cross-section through one of the lateral line scales to show how deep the groove is.

Line 236. “A few additional scales are interspersed…” What morphotype do these scales belong to?

Line 254. Parayunnanolepis is now by far the best-described placoderm body squamation. How do the authors rule out the attribution of these scales to other taxa?

Line 263. For clarity’s sake may be worth specifying that the sections made by Giels et al. were of is Yunnanolepis.

Line 272. In figure 6A it looks like there are several relatively large vascular openings in the scale e.g. left of "go1" without line, directly below go1 with line. Although these don't seem likely to be a thick middle layer like in the head armour they are worth commenting on. Is it a coincidence that they are in the largest scale?

Line 278. It should be stated what morphotypes the sectioned scales belong to.

Line 308. Also, in chondrichthyans? Living chondrichthyans' monodontode scales are arguably simpler than those of stem-group members such as acanthodians (e.g. Vernicomacanthus3).

Line 325. This again requires some phylogenetic clarity: using total-group descriptors osteostracans are jawed vertebrates. They are however the immediate outgroup to mandibulate jawed vertebrates.

Line 326. This is notably missing a reference to O'Shea et al. 2019, who conducted a detailed review of tremataspid scales over the whole body.

Line 326. This would be improved by at least mentioning other jawless stem-group jawed vertebrates for a complete discussion. There exist sections of body scales from heterostracans (Keating et al.), anaspids (Keating and Donoghue) and numerous thelodonts (e.g. Märss et al.)

Line 370. See comments above about alternative phylogenetic hypotheses.

Line 383. The references cited do not give a complete overview of stem-chondrichthyan scale morphologies. They include "classic" onion skin acanthodian scales (in Cheiracanthus, Trizeugacanthus, diplacanthids) and mongolepids etc, but omit the quite different areally growing scales of for example Parexus, ctenacanth-like scales like in Vernicomacanthus3 and monodontode scales in e.g. Lupopsyrus. These or equivalent examples should be added.

Line 391. This is true of the classic acanthodian scale morphology, but quite different ornaments are found on diverse stem-chondrichthyan scale forms (see above).

Line 396. See comments above about alternative phylogenetic hypotheses.

Line 401. I see where the authors are coming from but is this necessarily true? Palaeozoic chondrichthyans scales and head tesserae can have quite complex surface ornament but also lack a middle layer.

Line 402. Given that this is probably the most detailed-yet description of in situ stem-gnathostome squamation I think the discussion would benefit from even a brief comparison of the gross anatomy of squamations in early vertebrates. How do other articulated stem-gnathostome and early crown-group gnathostome squamations compare to Parayunnanolepis? Are they similarly heterosquamous? Do they have similar e.g. enlarged dorsal scales?

Line 421. Worth specifying here that the authors are talking about body scales (as opposed to the head skeleton).

Comments on figures

Figures are generally great and show everything described in lavish detail. I've made some suggestions for additions in the line-by-line comments above. Otherwise, some comments on the figures are below.

Figure 1. Figure 6E is a bit confusing as it's been flipped to make it anterior to the right. But (if I'm interpreting it correctly) this makes dorsal ventral and ventral dorsal. I think it would be clearer to rotate the panel 180 degrees so anterior is to the left, dorsal is up, and ventral is down.

Figure 1. The scale bars are a bit confusing. Does the scale on 1E apply to all panels 1D-I?

Figure 8. It would be easier to compare the scales if they had a consistent anterior direction, which I don't think they do.

Figure 8. As outlined above (line 326) I think the paper would benefit from more general comparisons to jawless gnathostomes. If so, they could be added to this figure.

Figure 8. I think "lose" and "gain" middle layer would be clearer than "presence" and "absence", which are really the states at the tips of the tree.

Figure 8. How did the authors arrive at the distribution of gained/lost characters? It strikes me that it would be just as parsimonious for a middle layer to be plesiomorphic for gnathostomes, and then to be lost four times in yunnanolepids, rhenanids, acanthothoracids, and chondrichthyans.

---

## [Author Response]

Reviewer #2 (Recommendations for the authors):In general, the article is well-written and provides clear and concise descriptions of the various morphotypes and squamation of the individual specimen; however, there are many sections where there are some grammatical or editorial mistakes. These include:Lines 32-34: How does this study demonstrate that yunnanolepidoid scales are different from the dermal plates of antiarchs, but also include yunnanolepidoids? The end of this sentence is a bit confusing.

Rephrased.

Line 45: I don't think the word "perfectly" should be used here as Reif, 1985 is just suggesting that different dermal denticles in sharks might correspond with different functions due to ecology or lifestyle. There hasn't been any explicit research showing evidence of this.

We deleted ‘perfectly’.

Lines 59-60: This sentence is confusing. The gnathal plates on placoderms can be extensively studied, not the other way around.

Rephrased.

Lines 68-69: The way the sentence is written currently suggests that P. xitunensis is known for preserving squamation. I think here you mean the "fossil" or "preserved specimen".

Rephrased.

Lines 81-83: This sentence doesn't make sense.

Deleted.

Line 103: What do you mean by "the size getting larger"? Increase in length, width?

Yes, the length and the width increase towards the posterior direction. We add “in length and width”.

Line 128: What changes the outline?

We deleted “changes the outline”.

Line 430: How big was this specimen? There is some evidence that dermal scales in sharks change throughout ontogeny so knowing the size of this specimen may help elucidate squamation growth if future individuals are recovered or examined.

We added “The post-thoracic body of *P. xitunensis* is about 3.8 mm in width, 6.7 mm in height, and 35.6 mm in length, and occupies approximately 58% of the length overall”.

Line 43: Jerve et al., 2017 isn't the actual article that differentiates the dermal skeleton into two different categories. I think this should be "Janvier P. Early Vertebrates. 1996".

Adopted.

Lines 54-56: Why are the citations here italicized?

We revised it.

Line 273: Do you mean "never" not ever? This is also done in Lines 318 and 393.

We deleted “ever” in all these three sentences.

Line 308: Could you give some examples of crown sarcopterygians and actinopterygians that you are referencing here?

We added the references.

Line 321: Should be "In this section" not "In this part".

Adopted.

Line 442: All of these abbreviations should be used in the text when the figure is referred to.

Adopted.

Line 87: Is the post-thoracic body indicated by the white region in Figure 1J? If so, this should be referenced here.

Yes, the white region represents the post-thoracic body. We regrouped the panels and quoted another figure here.

Line 93: Tubercles should be labeled in Figure 1.

Adopted.

Line 179: The lateral lamina is referenced with Figure 4C, E, but there is no lateral lamina in this figure.

Adopted. We added the label ‘ll.vs’ for the lateral lamina of the ventral scutes in the new Figure 5C, E.

Line 199: I think you mean supplemental figure 6, not 5.

Revised.

Line 326: The manuscript says that in tremataspid scales, the middle layer is well described; however, this is not depicted in Figure 8.

We modified the text and added another section image of a tremataspid scale in new Figure 9 (original Figure 8).

Lines 353-354: Again, the text says that Romundina bear a well-developed middle layer; however, this is not shown in Figure 8.

We explained it in the text as follows: “This histological difference in the *Romundina* scales probably results from individual disparity, as suggested by *Tremataspis mammillata* with reduction of the middle layer posteriorly across the squamation (O”Shea et al., 2019)”.

Line 401: Ornamentation is mentioned here but is not shown or labeled in Figure 8.

We added labels in the new Figure 9 (original Figure 8).

Line 426: This should actually be supplemental figure 5, not 6.

Revised.

Generally, the figures are really great and have a lot of information and detail that add to the manuscript; however, I think the authors should take some time to make sure the images can stand alone without needing the manuscript text for explanation. Additionally, there are some things specifically about the Figures themselves that I think could be changed to increase the readability of the manuscript:Line 713: Should read "(D-I) magnified images of rectangles in (A-C)".

Revised.

Figure 3 A and B: The lines that are separating the posterior flank area from the anterior flank area should be better described or labeled on the image. Also, the authors should point out here in the caption what the orange scales are.

Adopted.

Figure 7: The title should be "Evolution of squamation patterns of the caudal region in antiarchs".

The title now reads: "Evolution of squamation patterns of the post-thoracic region in antiarchs".

Figure 8: You should include what the view or plane is of the scales (or the two columns) we are looking at. Is the first column dorsal or lateral? Additionally, why is there a "?" in front of the M in the first row? It might be better if you change the colors of the presence and absence of the middle layer because currently, the color of the Middle layer bracket in column 2 is the same color as the bar representing a lack of a middle layer in the phylogeny. It should be consistent.

We added the explanation about the first columns, deleted ‘a’ in front of the M, and changed the colors.

Figure supplement 2: Line 802, should the Dorsal view letters be subscript "2" and not "1"?

Revised.

Figure supplement 3: How can the E scales be both anterior and posterior flank scales? Also, the label "H" is confusing. What does that point to?

We revised the figure caption and deleted “H”.

Figure supplement 5: Which morphotype are these scales? Are they all the same? Also, which plane are the rows showing?

Following suggestions, we added “V28643, V28645 and V28646 can be assigned to Morphotype 8; V28644 and V28647, probably from the posterior flank area, can be referred to Morphotype 9; V28642 can be referred to Morphotype 11”. We also added the explanation about the plane.

Reviewer #3 (Recommendations for the authors):I enjoyed reading this article, it's great to see people working on the posterior half of placoderms as opposed to their more popular business ends. I think that the articulated squamations of these things have untapped potential as a source of characters and information on vertebrate evolution which work like that presented here is necessary to access. Moreover, this data is linked to an animal commonly used in phylogenetic matrices targeting early vertebrates and I'm sure will be incorporated into future such work.The introduction outlines necessary background information clearly and succinctly. In terms of the analysis, CT scanning and thin sections are both very appropriate methods for investigating this thing's squamation, and the interpretation of this and the morphotypes proposed by the authors seem sensible. I think the discussion could range a little more broadly in terms of comparing the scales to those of other early vertebrates: more details are given below.My only issue with the interpretation of the data is in Figure 6A, where it seems to be there are some canals passing through the section. Why do the authors not consider this to be a middle layer? Even if they are confident it is not, this is worth discussing in the text. It seems to me that it's not impossible that the presence/absence of a middle layer can vary in a single taxon, depending on the size or the ornamentation of scales in different places. Dearden et al. 2021 (p22) suggest that "putative cancellous layer in placoderm scales could be an overinterpretation of the canals above the base in crown tubercles of placoderm scales (which also occur in stem-group chondrichthyan scales), rather than corresponding to the spongiosa or trabecular layer in placoderm dermal bone." which could be the case here?Given the tendency toward turmoil in stem-gnathostome relationships, I also think that it would benefit from considering alternative phylogenetic hypotheses for the phylogenetic position of antiarchs in the gnathostome stem group. While it's true that usually they are recovered in a fairly early-branching position in the gnathostome stem group, a recent analysis by Zhu et al. recovered them in quite a crownwards position. This is worth considering as an alternative when hypothesising about the evolution of vertebrate squamations. Or, if the authors do not agree with the phylogenetic proposition of that study, they should state why.Also, as noted in the figures section, the scheme of gain/absence in figure 8 is equally parsimonious to one where the middle layer is lost 4 times. What is the basis for this character optimisation?Another general point is that the authors repeatedly refer to mandibulate gnathostomes (i.e. gnathostomes with jaws) as "gnathostomes", for example when describing osteostracans as the outgroup to jawed vertebrates. Given that a total-group framework, which leads to numerous jawless vertebrates including osteostracans in the gnathostome stem group, has been widely adopted in early vertebrate palaeontology this can read a bit confusingly. I recommend that throughout, the authors are more terminologically precise: the use of e.g. "mandibulate stem-gnathostomes" etc might help.This dataset provides a good opportunity to compare the squamation of Parayunnanolepis more broadly than the authors do at present. If antiarchs are indeed the earliest branching mandibulate jawed vertebrates then it is worth comparing their squamation beyond the immediate outgroup (osteostracans) to articulated squamations in heterostracans, theodonts, anaspids, galeaspids (if any exist?).My line by line comments follow:AbstractLine 16: "the most basal jawed vertebrates". Potentially misleading as the word basal is ill-defined. Moreover, even if it is taken to mean the earliest branching jawed vertebrates, "ostracoderms" are the most "basal" jawed vertebrates. Recommend rewriting using terminology like "earliest branching mandibulate jawed vertebrates".

We changed it into "earliest branching jawed vertebrates", following the reviewer’s suggestion. “mandibulate” is redundant with “jawed”. “Ostracoderms”, as jawless fishes, are the most basal stem-gnathostomes or the most basal gnathostomes (in sense of ‘total group gnathostomes’), rather than the most basal jawed vertebrates. ‘Jawed vertebrates’ are the vertebrates with jaws. As such, they are same as the mandibulate vertebrates in sense of the reviewer.

Line 42: "as a homogenous squamation across the animal". This isn't strictly speaking true: many acanthodians have surprisingly heterogenous squamations with (e.g. Ptomacanthus Brazeau 2012, Vernicomacanthus Dearden et al. 2021), as do living chondrichthyans (e.g. Naylor et al. 2021) All would be considered micromeric.

We deleted “homogenous”.

Line 45 Does anything in nature correspond "perfectly"?

We deleted “perfectly”.

Line 49 Like basal, "upper" is terminologically unclear. Using more precise terminology like "more crownwards grade" or "a grade with a closer relationship to the crown group" would be clearer.

We changed “upper” to “crownward”.

Line 64 "at the root of jawed vertebrates" See comments on line 16. Also, see the general comment above on alternative hypotheses of relationships.

See our response above on the definition of ‘jawed vertebrates’.

Line 86 I find the authors' scale morphotypes convincing, but the paper would benefit from an additional figure showing all 13 morphotypes side by side. This will help readers get a quick overview and sense of the variation and morphology of the morphotypes.

Adopted.

Line 90. I think in the summary descriptions it is also worth stating what characters that readers might expect in early vertebrate scales are absent. For example, are there any basal pores or neck canals? I understand that there are not, but it is worth stating their absence as they are potentially important characters of early vertebrate body scales.

We added “Basal pores are less developed. Scales are relatively flat, and barely bear the neck structure” in the text.

Line 125 Morphotype 4 could do with a more detailed description of the morphology to bring it in line with the other morphotype descriptions.

Adopted.

Line 204. "…predorsal area shows the large disparity…" Should read "largest"?

We changed “large” to “largest”.

Line 214. It would be useful to figure a cross-section through one of the lateral line scales to show how deep the groove is.

Adopted.

Line 236. "A few additional scales are interspersed…" What morphotype do these scales belong to?

We added “A few additional scales….. resembling the surrounding scales in morphotypes”.

Line 254. Parayunnanolepis is now by far the best-described placoderm body squamation. How do the authors rule out the attribution of these scales to other taxa?

We added more explanation here. In fact, we just assigned them as “yunnanolepidoid scales” instead of “*Parayunnanolepis* scale”.

Line 263. For clarity's sake may be worth specifying that the sections made by Giels et al. were of is Yunnanolepis

Adopted. The text now reads: “Thus, the histology of yunnanolepidoid scales is remarkably different from that of their dermal bony plates, which exhibits a three-layered structure including a thick cancellous spongiosa in the middle as in *Yunnanolepis* and *Chuchinolepis* (Giles et al., 2013). Significantly, the dermal skeletal differentiation across the dermal shield and scales with respect to the middle layer, is also the case for heterostracans and osteostracans (Denison, 1947; O'Shea et al., 2019).”

Line 272. In figure 6A it looks like there are several relatively large vascular openings in the scale e.g. left of "go1" without line, directly below go1 with line. Although these don't seem likely to be a thick middle layer like in the head armour they are worth commenting on. Is it a coincidence that they are in the largest scale?

We added “The vasculature might be supplied by occasional ascending canals (*Figure 7A*, vc) from the basal layer, which opened to the surface but was buried later by younger odontodes. In this case, the middle/spongiosa layer may be poorly developed or absent in certain morphotypes” in the text, and labelled “vc” (vascular canal) in the figure. We will investigate the development of the middle layer in an individual in the follow-up works.

Line 278. It should be stated what morphotypes the sectioned scales belong to.

See our response above for “Figure supplement 5”.

Line 308. Also, in chondrichthyans? Living chondrichthyans' monodontode scales are arguably simpler than those of stem-group members such as acanthodians (e.g. Vernicomacanthus3).

See our response above for “Figure supplement 5”.

Line 325. This again requires some phylogenetic clarity: using total-group descriptors osteostracans are jawed vertebrates. They are however the immediate outgroup to mandibulate jawed vertebrates.

See our response above on the definition of ‘jawed vertebrates’. Osteostracans are gnathostomes or stem gnathostomes using the total group concept, rather than jawed vertebrates or mandibulate vertebrates.

Line 326. This is notably missing a reference to O'Shea et al. 2019, who conducted a detailed review of tremataspid scales over the whole body.

We added this reference here.

Line 326. This would be improved by at least mentioning other jawless stem-group jawed vertebrates for a complete discussion. There exist sections of body scales from heterostracans (Keating et al.), anaspids (Keating and Donoghue) and numerous thelodonts (e.g. Märss et al.).

Adopted.

Line 370. See comments above about alternative phylogenetic hypotheses.

We added “Alternatively, under the recent phylogenetic hypothesis by Zhu et al. (2021), the most parsimonious solution is that the well-developed middle layer independently evolved four times in petalichthyids + arthrodires, brindabellaspids, euantiarchs and osteichthyans”.

Line 383. The references cited do not give a complete overview of stem-chondrichthyan scale morphologies. They include "classic" onion skin acanthodian scales (in Cheiracanthus, Trizeugacanthus, diplacanthids) and mongolepids etc, but omit the quite different areally growing scales of for example Parexus, ctenacanth-like scales like in Vernicomacanthus3 and monodontode scales in e.g. Lupopsyrus. These or equivalent examples should be added.

We added more references concerning other types of acanthodian scales.

Line 391. This is true of the classic acanthodian scale morphology, but quite different ornaments are found on diverse stem-chondrichthyan scale forms (see above).

We provided more details here.

Line 396. See comments above about alternative phylogenetic hypotheses.

See our response above for “Line 370”.

Line 401. I see where the authors are coming from but is this necessarily true? Palaeozoic chondrichthyans scales and head tesserae can have quite complex surface ornament but also lack a middle layer.

We deleted the sentence.

Line 402. Given that this is probably the most detailed-yet description of in situ stem-gnathostome squamation I think the discussion would benefit from even a brief comparison of the gross anatomy of squamations in early vertebrates. How do other articulated stem-gnathostome and early crown-group gnathostome squamations compare to Parayunnanolepis? Are they similarly heterosquamous? Do they have similar e.g. enlarged dorsal scales?

We thank the reviewer’s suggestions and consider adopting it in the follow-up studies.

Line 421. Worth specifying here that the authors are talking about body scales (as opposed to the head skeleton).

We deleted the sentence.

Comments on figuresFigures are generally great and show everything described in lavish detail. I've made some suggestions for additions in the line-by-line comments above. Otherwise, some comments on the figures are below.Figure 1. Figure 6E is a bit confusing as it's been flipped to make it anterior to the right. But (if I'm interpreting it correctly) this makes dorsal ventral and ventral dorsal. I think it would be clearer to rotate the panel 180 degrees so anterior is to the left, dorsal is up, and ventral is down.Figure 1. The scale bars are a bit confusing. Does the scale on 1E apply to all panels 1D-I?

Yes, the scale on 1E apply to all panels 1D-I. We added explanations about the figure panels 1D–I and the scale bar.

Figure 8. It would be easier to compare the scales if they had a consistent anterior direction, which I don't think they do.

Adopted.

Figure 8. As outlined above (line 326) I think the paper would benefit from more general comparisons to jawless gnathostomes. If so, they could be added to this figure.

Thanks for the suggestions and we consider making broad comparisons in the follow-up studies.

Figure 8. I think "lose" and "gain" middle layer would be clearer than "presence" and "absence", which are really the states at the tips of the tree.

We thought “presence/absence” is consistent with the whole text.

Figure 8. How did the authors arrive at the distribution of gained/lost characters? It strikes me that it would be just as parsimonious for a middle layer to be plesiomorphic for gnathostomes, and then to be lost four times in yunnanolepids, rhenanids, acanthothoracids, and chondrichthyans.

We have discussed in the text that the bipartite scale in yunnanolepidoids, rhenanids, and acanthothoracids is more likely to represent the plesiomorphic condition for jawed vertebrates.